# Effect of Minor Er Additions on the Microstructures and Mechanical Properties of Cast Al-Cu-Mg-Ag Alloys

**DOI:** 10.3390/ma14154212

**Published:** 2021-07-28

**Authors:** Haoyu Xie, Juangang Zhao, Jing Cao, Lei Luo, Shuai Guo, Linnan Ou, Zhiyi Liu, Song Bai

**Affiliations:** 1School of Material Science and Engineering, Central South University, Changsha 410083, China; 18005667858@163.com (H.X.); 163101014@csu.edu.cn (J.Z.); jingcao@csu.edu.cn (J.C.); luolei1@csu.edu.cn (L.L.); 193111034@csu.edu.cn (S.G.); 193112169@csu.edu.cn (L.O.); 2Key Laboratory of Nonferrous Metal Materials Science and Engineering, Central South University, Changsha 410083, China; 3The National Key Laboratory of Science and Technology on High-Strength Structural Materials, Central South University, Changsha 410083, China

**Keywords:** Al-Cu-Mg-Ag alloy, precipitation, Ω phase, θ′ phase, Er

## Abstract

The microstructures and mechanical properties of novel cast Al-Cu-Mg-Ag alloys with and without minor additions of Er (0.09 and 0.2 wt %) are investigated by Vickers hardness tests, tensile tests, optical metallographic examination, scanning electron microscopy (SEM), transmission electron microscopy (TEM), and X-ray diffraction (XRD). The results reveal that the Er addition decreases the hardness value of peak-aged Al-Cu-Mg-Ag alloy but has little influence on the time required for achieving the peak aging condition. Meanwhile, the Ω phase is suppressed in Er-added alloys, leading to a lower tensile strength at room temperature, which causes the (Mg, Ag, Er, V, Ti)-rich phase in the matrix in Er-added alloys. This blocky phase consumes available Mg and Ag atoms for Ω nucleation, leading to the low number density of Ω plates. The strength properties of Er-added alloys at 300 °C are found to be enhanced, which benefits from the pinning effect of the Al_8_Cu_4_Er phase on grain boundaries. Meanwhile, the brittle fracture of Er-added alloys at room temperature is directly associated with the Al_8_Cu_4_Er phase and the blocky (Mg, Ag, Er, V, Ti)-rich phase, which acts as the source of microcracks during deformation. In addition, no obvious grain refinement effect can be observed in Er-added alloys.

## 1. Introduction

Being a new type of high-strength and heat-resistant aluminum alloy, the Al-Cu-Mg-Ag alloy has been widely used in the field of aeronautics due to its excellent creep resistance and fatigue resistance at elevated temperatures [1,2,3]. These properties are mainly attributed to the particular precipitation in the alloy–Ω phase. The addition of Mg and Ag elements to the Al-Cu-based alloy has been documented to provide a favorable site for nucleation of the Ω phase on {111}_α_ planes [4]. Moreover, the θ′ phase is also formed in the alloy, competing with the Ω phase for Cu atoms in the Al matrix during the aging process [5]. In contrast, the Ω phase not only exhibits high strength and excellent resistance to coarsening, but also reduces the coherency misfit strain and provides relatively low energy for coherent boundaries at elevated temperature of 250 °C [6,7,8]. However, the coarsening rate of Ω phase accelerates when the temperature further increases to 300 °C, limiting the use of the Al-Cu-Mg-Ag alloy [9,10].

Development through modification of the elemental constitution is an economical approach in application. The addition of rare-earth elements can effectively enhance the mechanical properties of aluminum alloy. Being new micro-alloying elements, the potential value of rare-earth elements such as Ce, Yb, and Sc has attracted significant academic scrutiny. According to the study conducted by Xiao et al., the addition of 0.45 wt % Ce induces the denser and finer Ω and θ′ phase, which boosted the yield strength of Al-5.3Cu-0.8Mg-0.6Ag alloy [11]. Nevertheless, Li et al. suggested a contrary view proving that 0.23 wt % Ce reduces the tensile properties of Al-4.66Cu-0.39Mg-0.48Ag alloy [12]. Moreover, Xiao et al.’s work demonstrates that the stability of the Ω phase is enhanced by the addition of 0.35 wt % Yb, improving the mechanical properties of Al-5.0Cu-1.0Mg-0.5Ag alloy at elevated temperatures [13]. As opposed to Ce and Yb, Sc is claimed as the most effective strengthening element in aluminum alloys, which is associated with its grain refinement and dynamic recrystallization controlling effect due to the high solid solubility in the α-Al matrix. For example, in Xue et al.’s work, the grain size of the Al-2.4Cu alloy is obviously refined after adding 0.1 wt % Sc [14]. Li et al.’s work also demonstrates that Al-4.12Cu-0.48Mg-0.53Ag-0.11Sc shows better thermal stability compared with the Sc-free alloy due to the formation of Al_3_Sc phases [15]. Furthermore, a previous study verified that a minority of the Sc exists as the form of Al_8−x_Cu_4+x_Sc phase and most of the Sc forms Al_3_(Sc, Zr) particles in Al-5.6Cu-0.72Mg-0.5Ag alloy, both of which are helpful to the mechanical properties at high temperatures of the alloy [16]. However, these rare-earth elements are limited in use in mass production because of their high price.

Erbium has garnered significant interest from researchers as an alternative element due to its low price. Furthermore, Er also acts as a grain refiner in Al-based alloys. Moreover, the Al_3_Er phase, which possesses the same L1_2_ structure as the Al_3_Sc phase precipitated in aluminum alloy, can also be deemed a major strengthening phase. Based on Pan et al.’s research, the mechanical properties of Al-3Cu alloy are enhanced by the formation of Al_3_Er particles, also providing numerous nucleation sites for the θ′ phase [17]. Moreover, according to Li et al.’s work, the addition of minor Er favors a significant refinement in dendritic substructure in the Al-4.6Cu-0.39Mg-0.48Ag alloy [18]. However, Bai et al. found a coarser grain structure in the Al-6.23Cu-0.61Mg-0.47Ag-0.23Er alloy compared with the Er-free alloy, and the tensile properties also deteriorated [19].

As a potential material serving at elevated temperatures, more stringent requirements are set regarding aluminum alloys. Although the potential effects of Er on Al-based alloys’ grain structure and microalloying behavior have been thoroughly documented, most research still focuses on deformed alloys such as hot-rolled sheets or extruded pipes [20,21,22,23], which undoubtedly increase the production cost, particularly considering complex work pieces. The design of a novel cast Al-Cu-Mg-Ag alloy is, therefore, a matter of pressing urgency. This study mainly aims to explain the association between the tensile properties and the microstructures in order to design the optimal Er content as well as improving the mechanical properties of the Al-Cu-Mg-Ag alloy at 300 °C.

## 2. Experimental Details

All of the experimental alloys were cast using traditional casting methods. The chemical compositions of the studied alloys are provided in Table 1. All tested samples were solution-treated in a salt bath furnace at 530 °C for 12 h, then artificial aged at 165 °C for various time periods after water quenching.

The peak aged state was determined by Vickers hardness measurements applied at load of 3 kg for 15 s, and each hardness value was measured with at least five measurements from five samples.

The “dog-bone”-type tensile tests specimens were machined according to ISO 783-1999, then carried out at room temperature on an Instron-MTS810 machine (SINOTEST, Changchun, China) and at 200 °C and 300 °C on a RWS50 machine (MTS, Minneapolis, MN, USA) with a strain rate of 10^−3^ s^−1^ to obtain the tensile properties. For each alloy, three specimens were repeatedly experimented to ensure the accuracy of the data.

The optical microscope (OM), X-ray diffraction (XRD), scanning electron microscope (SEM), and transmission electron microscope (TEM) were applied to understand and clarify the discrepancies among all studied alloys. The metallograph was monitored by Caikon DMM-900C OM machine (Caikon, Shanghai, China) and the grain size was measured using Image J v1.8.0 software (Bethesda, MD, USA), with at least 500 grains of low magnification for each alloy. The main phases in different alloys were performed on a Rigaku D/Max 2500PC X-ray diffractometer (Bruker, Karlsruhe, Germany). The morphologies and the chemical compositions of different intermetallics were examined by FEI Quanta 200 SEM machine (FEI, Hillsboro, OR, USA). The differences of the precipitates in the three alloys were characterized by a Tecnai G^2^ 20 TEM machine (FEI, Hillsboro, OR, USA). All TEM samples were electro-polished in a solution of 70% methanol and 30% nitric acid at approximately −20 °C. The average length, thickness, and number density of Ω and θ′ precipitates were measured by means of Image J software with at least 2000 precipitates from five different grains.

The method of Underwood was used to calculate the volume fraction of Ω and θ′ precipitates [24,25]. The volume fraction ($f_{v}$) of Ω plates and θ′ is calculated by:(1)$$f_{v-\Omega }=\frac{\pi {\left(\bar{d}\right)}^{2}\bar{t}}{4}\cdot \frac{N\left(1+\frac{T+\bar{d}}{\sqrt{A}}\right)}{A\left(T+d\right)}$$
(2)$$f_{v-\theta^{\prime }}=\frac{\pi {\left(\bar{d}\right)}^{2}\bar{t}}{4}\cdot \frac{N\left(1+\frac{2\left(T+\bar{d}\right)}{\sqrt{A}}\right)}{A\left(T+d\right)}$$

And the number density (N) of Ω and θ′ precipitate is calculated by:(3)$$N=4\frac{V}{{\pi \bar{t}\bar{d}}^{2}}$$
where A is the corresponding area of the image, $\bar{d}$ and $\bar{t}$ represent the average plate diameter and thickness of the plate-shaped precipitates, respectively; T is the foil thickness obtained by the convergent beam electron diffraction (CBED) method by using Kossel–Mollenstedt fringe spacing [24,26,27].

## 3. Results

### 3.1. Age-Hardening Curves

Figure 1 shows the plot of age-hardening curves of investigated alloys with and without Er additions at 165 °C. It is found that in Figure 1, the trend of all curves is basically the same with a rapid increase and then a gradual decline. It is worth noting that minor Er addition has little effect on altering the time to peak aging condition with all studied alloys reaching peak aging condition (T6 state) for 14 h. However, the maximum hardness of 141.8 ± 3.5 HV for Er-free alloy decreases to 138.1 ± 2.7 HV in 0.09Er alloy and finally drops to 137.9 ± 2.8 HV in 0.20Er alloy. After the peak aging stage, the Vickers hardness of Er-free alloy is still superior to that of 0.09Er and 0.20Er alloys.

### 3.2. The Tensile Properties

Tensile properties of all T6 tempered alloys at different temperatures are revealed in Figure 2. With the Er content increasing from 0 wt % to 0.20 wt %, the tensile strength at room temperature decreases from 472 MPa to 452 MPa and ultimately drops to 438 MPa. On the contrary, the results of the tensile strength of the studied alloys at 300 °C are totally different, with the value of 195, 222, and 201 MPa for Er-free, 0.09Er, and 0.20Er alloy, respectively. In addition, a similar tendency is also found in yield strength of the three studied alloys with the increasing Er content. It is found that the highest yield strength of 431 MPa can be obtained in the Er-free alloy at ambient temperature. While under high temperature condition, the 0.09Er alloy shows the best yield strength, with a value of 220 MPa at 300 °C, which is much higher than that of the Er-free alloy (194 MPa) and the 0.20Er alloy (200 MPa).

In contrast, the reduction of elongation is detected with the addition of Er. At room temperature, the Er-free alloy owns the uppermost elongation about 5.3% after fracture, while the elongation value of 0.09Er and 0.20Er alloy drops to about 3.9%. Furthermore, the elongation among all alloys is always found to be above 9% at 300 °C. It can be concluded that small Er addition degrades the tensile properties of peak-aged Al-Cu-Mg-Ag alloys at ambient temperature but increases the tensile properties at elevated temperature.

### 3.3. Grain Structures

The metallographic photos of the as-cast alloys are shown in Figure 3. The as-cast Er-free alloy is predominated by a large amount of coarse dendritic structures and few fine equiaxed grains of about 86.7 μm in diameter. Whereas, it is worth noting that with the addition of Er element, the grain size did not change much. A small number of typical dendritic substructures as well as a large sort of fine equiaxed grains are still maintained with the size about 82.3 and 85.2 μm in Figure 3b,c, indicating that the effect of grain refinement of Er in this work is not obvious. In addition, a few coarse second-phase particles (marked by black arrows) appear in the grains of the alloy with Er addition.

### 3.4. XRD Analysis

The XRD patterns of three studied alloys after peak aging treatment are presented in Figure 4. For Er-free alloy, the XRD patterns only included peaks of α-Al and θ-Al_2_Cu [26]. While in the XRD patterns of 0.09Er alloy, the intensity of diffraction peak of Al_8_Cu_4_Er phase is not obvious since the volume fraction of Al_8_Cu_4_Er phase must be very low (<1%), which is even lower than the minimum detectable level of X-ray diffractometer. For 0.20Er alloy, extra peaks of Al_8_Cu_4_Er phase are observed, and the peak intensity of Al_8_Cu_4_Er phase increases significantly with the increase in Er content. However, after solution treatment, there is no significant difference in the peak intensity of the Al_8_Cu_4_Er phase in Er-added alloys, which proves its thermal stability. The peak intensity of the Al_2_Cu phase decreases but does not disappear completely, which is mainly caused by the fact that the Cu content in all tested alloys is higher than the maximum solid solubility in Al.

### 3.5. SEM Observation

The backscattered electron images (BSE) of the as-cast experimental alloys are shown in Figure 5. It is obvious that all studied alloys are predominated by reticulated dendritic α-Al and continuous intergranular primary phase. According to the results of EDS analysis in Table 2, the coarse gray phase in Er-free and Er-added alloys is believed to be Al_2_Cu phase (point 1, 4, and 8) in view of the atomic ratio of Al and Cu with 2:1, which is consistent with the Al-Cu-Mg diagram [28,29]. Additionally, the impurity Fe in the alloy segregates with Cu and Mn forms the AlCuMnFe phase in the matrix (point 2). However, a different brighter phase is detected at the grain boundary with the addition of Er. Points 3 and 6 indicate the bright particles are enriched with Cu and Er with the mole ratio of 4:1, which could be considered as Al_8_Cu_4_Er phase according to the Al-Cu-Er phase diagram [30] and XRD patterns in Figure 4. Meanwhile, another blocky phase consisting of several elements is discovered in the Er-added alloys (points 5 and 7), which is marked as the (Mg, Ag, Er, V, Ti)-rich phase in this paper. What is noteworthy is that the volume fraction of the thick, blocky intermetallics increases from 1.03% to 2.27% as the Er content increases from 0.09% to 0.20%.

Figure 6 shows the BSE images of different alloys after T6 heat treatment. Evidently, after the solution treatment, most second phase dissolves into the matrix, and second-phase particles along the grain boundaries change from continuous to discontinuous and thinner. However, due to the content of Cu in all studied alloys being far beyond the limit of solubility in Al, a small number of Al_2_Cu phase is still retained after solution treatment, which can be confirmed by EDS analysis of points 1, 4, and 7 in Table 3. Additionally, according to points 2 and 6 in Figure 6, it should be pointed out that Al_8_Cu_4_Er phase still remains in Er-added alloys after solution treatment. Similarly, coarse (Mg, Ag, Er, V, Ti)-rich phase was also not dissolved in the Al matrix, which could be proved by points 3, 5, and 8 in Figure 6 as well as EDS mapping in Figure 7. In addition, it is worth noting that the volume fraction of the blocky (Mg, Ag, Er, V, Ti)-rich phases did not change significantly compared with that before solution, which is sufficient to demonstrate its thermal stability at 300 °C.

### 3.6. TEM Observation

#### 3.6.1. The Ω Phase and θ′ Phase

The TEM observation near the <110>_α_ orientation in all experimental alloys is illustrated in Figure 8. It can be seen that the major strengthening phase in all alloys is the Ω phase, which corresponds to the diffraction spots of 1/3 and 2/3 {220}_α_ positions. According to Table 4, the number density of Ω plates in the Er-free alloy is estimated to be 27.6 ± 10.3 × 10^3^/μm^3^, about 24% and 44% higher than that (21.1 ± 9.1 × 10^3^/μm^3^ and 15.4 ± 1.1 × 10^3^/μm^3^) of the 0.09Er and 0.20Er alloy, respectively. Compared with the average diameter of Ω plates in Er-free alloy (42.3 ± 12.4 nm), the average diameter of it in 0.09Er and 0.20Er alloy is almost the same (37.78 ± 9.5 nm and 37.5 ± 6.3 nm). Moreover, according to statistics, the addition of Er has little effect on altering the thickness of Ω plates, and the thickness of the Ω phase for all is basically maintained at about 2.5 nm.

In addition to the dense Ω phase, the alloys are accompanied by a large amount of θ′ phase. As shown in Figure 9a, there is only a sparse θ′ phase (marked by a black arrow) in the Er-free alloy. On the contrary, with the addition of the Er element, the θ′ phase in the matrix obviously increases as depicted in Figure 9b,c, which can be proved by the streaks at 1/2 {220}_α_ positions. In order to further verify the influence of Er on θ′ precipitates, the examination along <100>_α_ orientation is also chosen for analysis. The presence of dense θ′ precipitates can also be demonstrated by the reflections of 1/2 {220}_α_ along the <100>_α_ orientation in Figure 9b,c (marked as white arrows). According to Table 4, the number density of the θ′ phase in the Er-free alloy is estimated to be 2.65 ± 2.1 × 10^3^/μm^3^, which is absolutely lower than that of the 0.09Er (4.3 ± 3.1 × 10^3^/μm^3^) and 0.20Er (4.13 ± 6.7 × 10^3^/μm^3^) alloys, respectively. Furthermore, the average length of the θ′ phase in the Er-free alloy is about 32.7 ± 10.1 nm, quite different from that of the 0.09Er (36.8 ± 15.4 nm) and 0.20Er alloys (35.4 ± 16.3 nm).

TEM results showing how Er affects Ω and θ′ phases are presented in Figure 10. The distributions of length of Ω plates are presented in Figure 10a, which are similar in all alloys. However, the major average length of the Ω phase of the Er-free alloy is between 30 nm to 50 nm, while the length of the Ω plates of the Er-added alloys tends to be more even and shorter.

What is noteworthy is that the length distribution of θ′ precipitates in the alloy with and without Er is slightly different. θ′ precipitate lengths ranging from 20.0 nm to 30.0 nm are the majority of the population in the Er-free alloy, while lengths with a range from 30 nm to 40 nm dominate in Er-added alloys.

#### 3.6.2. Al_8_Cu_4_Er Phase

Based on the results of the tensile test and SEM images, it is necessary to reveal the microalloying behavior of erbium in Al-Cu-Mg-Ag alloys. Therefore, STEM-HAADF is specifically employed to observe the microstructures of the 0.09Er alloy in the T6 state after the tensile test at 300 °C. Based on Figure 11, the segregation of Cu and Er atoms appears at the grain boundary and according to the XRD results in Figure 4 it can be determined that Er tends to distribute to the grain boundaries and form the ternary Al_8_Cu_4_Er phase. It is also noteworthy that the Al_8_Cu_4_Er phase can still exist stably after a short period of time at 300 °C.

### 3.7. Fractography

Figure 12 shows SEM images of tensile fractures of three tested alloys at peak-aged state at a temperature of 25 °C. The selected positions enclosed by red boxes are magnified in Figure 12b,e,h, respectively. It is found that all alloys exhibit a typical brittle fracture pattern. It can be seen that there are some typical casting defects (marked as red circles in Figure 12c,f,i) in all alloys. Furthermore, the fracture surface of the Er-free alloy is covered by continuous intercrystalline compounds, which can be confirmed as coarse Al_2_Cu particles according to the EDS analysis results, whereas, the proportion of large Al_2_Cu is much lower than that of the Er-free alloy. In addition, the fracture surface of the 0.09Er and 0.20Er alloys contain more micro-voids and deeper tear ridges (marked by yellow arrows). Except for the continuous Al_2_Cu phase along the grain boundaries, two other phases are found at the fractures of the 0.09Er and 0.20Er alloys. One phase exists inside the dimples and can be identified as Al_8_Cu_4_Er phase according to the EDS analysis. The other is found in the matrix, and this intermetallic matrix contains Ag, Er, Ti, and V, which could be inferred from the above analysis and determined as the (Mg, Ag, Er, V, Ti)-rich phase.

## 4. Discussion

### 4.1. Microstructure Evolution with Er Addition

Generally, the addition of minor rare earth elements can refine the grain size, especially for Er, which can be used as an effective grain refiner in aluminum alloys [31,32,33]. However, the as-cast optical micrographs in Figure 3 indicate that Er-induced grain refinement is not significant. According to [34,35], the lack of obvious grain refinements may be explained from using other effective grain refiners such as Ti and B in this work, so that the grain refining effect of the Er element is not very obvious.

SEM results in this work clearly reveal how Er affects the overall morphology and distribution of residue particles in all alloys. Based on the SEM images in Figure 5 and Figure 6, the blocky (Mg, Ag, Er, V, Ti)-rich phase is formed in 0.09Er and 0.20Er alloys as they solidify, and the phase is not eliminated after solution treatment. It is noteworthy that the phase is rich in Mg and Ag elements, which are the key elements for the formation of the precursor of Ω phase–Mg-Ag co-clusters.

In [36,37,38], Mg-Ag co-clusters play an important role in nucleation and growth of the Ω phase. Bai et al.’s results demonstrate that the Ω phase is promoted as the Ag content increases. This is because during the early aging state, an Mg atom can be easily captured by Ag atoms, then the co-segregation of Mg and Ag atoms tends to form Mg-Ag co-clusters [39,40]. After solution treatment, a high density of Mg-Ag co-clusters are formed rapidly due to the strong tendency of the trapping effect of Ag atoms on Mg atoms in Al-Cu-Mg-Ag alloys. Then, during the aging process, the Mg-Ag co-clusters gradually grow by incorporating Cu solutes continuously and thus result in the Ω phase [38]. According to the EDS analysis of the (Mg, Ag, Er, V, Ti)-rich phase in Figure 5, Figure 6 and Figure 7, a fair number of Mg and Ag atoms segregate to these blocky second-phases, resulting in insufficient Ag and Mg atoms dissolving into the α-Al matrix and finally leading to a significant decrease in the number of Ω phases in Er-added alloys.

On the contrary, the density of the θ′ phase in the Er-free alloy is much less than that in the 0.09Er and 0.20Er alloys. In general, the formation and the growth of the Ω and θ′ phases in Al-Cu-Mg-Ag alloys compete with each other by consuming Cu atoms in the matrix [41,42]. Thus, under the premise that the Ω phase is inhibited, the slightly oversaturated Cu atoms in the experimental alloy tend to form the GP zones preferentially and ensure the precipitation of the θ′ phase at 165 °C [43,44]. That is, though the presence of a mass of Al_8_Cu_4_Er intermetallics in the 0.09Er and 0.20Er alloys, which consume a great deal of Cu solutes, there is still sufficient Cu solutes for the precipitation of the θ′ phase. According to quantitative calculations, only 0.15 wt % of Cu will be consumed even if the content of 0.1 wt % Er is used to form Al_8_Cu_4_Er, let alone the Er atoms that have been dissolved in the matrix and segregated at the (Mg, Ag, Er, V, Ti)-rich phase. Therefore, 0.09Er and 0.20Er alloys consume up to 0.15 wt % and 0.30 wt % Cu to form Al_8_Cu_4_Er phase. So even though all Er atoms prefer to form Al_8_Cu_4_Er particles, the available Cu remaining (5.77 wt % in the 0.09Er alloy and 5.65 wt % in the 0.20Er alloy) for θ′ precipitation will still be sufficient. Consequently, the nucleation of θ′ precipitates will be accelerated owing to the sufficient Cu atoms in the 0.09Er and 0.20Er alloys.

### 4.2. Relationship between Microstructures and Mechanical Properties

As an age-hardenable aluminum alloy, the ultimate microstructures and corresponding mechanical properties of Al-Cu-Mg-Ag alloys are controlled by the nucleation and growth of the Ω and θ′ phases. According to the TEM characterization in Figure 8 and Figure 9 and Table 4, the number density of the Ω phase in the Er-free alloy is much higher than that of the 0.09Er and 0.20Er alloys while the number density of the θ′ phase in the Er-added alloys increases with the increase in Er content. The significant difference among these alloys demonstrates that the precipitation of the Ω phase in Er-added alloys is limited but the formation of the θ′ phase is promoted after Er addition. Therefore, it is necessary to analyze the contributions of these precipitations to the yield strength in all alloys. For the three studied alloys, it can be concluded that the yield strength is related to the density of Ω and the θ′ interactions with dislocations. According to [45], the contribution of these two different phases to the yield strength can be calculated as follows:(4)Δσp−Ω=0.12MGbd¯t¯·[fv12+0.7d¯t¯fv+0.12d¯t¯fv12]·ln0.079d¯r0
(5)$$\Delta \sigma_{p-\theta^{\prime }}=\frac{\mathrm{MGb}}{2\pi \sqrt{1-v}}\frac{1}{1.23\times 0.931\sqrt{\frac{0.306\pi \bar{d}\bar{t}}{f_{v}}-\frac{\pi \bar{d}}{8}-1.061\bar{t}}}\ln\left(\frac{0.981\sqrt{\bar{d}\bar{t}}}{b}\right)$$
where M is the Taylor factor, which equals 3.06. G, b, and υ represent the shear modulus of the matrix (27.8 GPa), magnitude of the Burgers vector (0.286 nm), and Poisson’s ratio for Al (0.345), respectively [25,46]. In addition to this, $\bar{d}$ is the average plate diameter, $\bar{t}$ is the average plate thickness, $f_{v}$ is the volume fraction of precipitates, $r_{0}$ represents the radius of the dislocation that by-passes the precipitates, which is equal to b.

Then, the total contribution of the Ω and θ′ phases to the yield strength of the alloys can be calculated as follows:(6)$$\Delta \sigma_{=}\Delta \sigma_{p-\Omega }+\Delta \sigma_{p-\theta^{\prime }}$$

Based on Equation (6), the overall strengthening contributions of different strengthening mechanisms to Er-free, 0.09Er, and 0.20Er alloys obtained by theoretical calculation are 360 MPa, 337 MPa, and 330 MPa. These are consistent with the tendency obtained by the experiments depicted in Figure 2a, indicating that the additions of Er have a deterioration effect on the mechanical properties of the Al-Cu-Mg-Ag alloys at room temperature.

Meanwhile, as a coarse second-phase in the 0.09Er and 0.20Er alloys, the (Mg, Ag, Er, V, Ti)-rich phase often acts as the origin of cracks during the tensile process. In general, the brittle fracture of an alloy is caused by the initiation, connection, and growth of microvoids near the coarse second-phase. Furthermore, the typical casting defects in cast alloys also provide favorable nucleation sites for microcracks in the alloy. Figure 12c illustrates that the fracture behavior of the Er-free alloy is controlled by intergranular semi-continuous Al_2_Cu, while the Al_8_Cu_4_Er phase and the blocky (Mg, Ag, Er, V, Ti)-rich phase controlled the fracture behavior of the Er-added alloy at 25 °C in addition to the Al_2_Cu phase. During the tensile process at room temperature, the secondary cracks are initiated near the Al_8_Cu_4_Er phase and the (Mg, Ag, Er, V, Ti)-rich phase after the separation of these Er-rich intermetallics from the grain boundaries and α-Al matrix (marked as blue arrows in Figure 12). On the other hand, some Al_2_Cu phases are torn under continuous stress, resulting in more secondary microcracks (marked as red arrows in Figure 12) at the same time. Finally, as the cracks are connected with each other, they expand and lead to the fracture of the alloys, which is consistent with the drop of tensile properties of Er-added alloys in Figure 2a.

When the tensile temperature rises to 300 °C, the tensile properties of the alloys with Er addition are improved compared with the Er-free alloy as shown in Figure 2b, which may be caused by the Al_8_Cu_4_Er phase at the grain boundaries of the 0.09Er and 0.20Er alloys. Compared with room temperature, the strength of the grain boundary is weaker than that of the matrix at the elevated temperature. Figure 11 also verified that the Al_8_Cu_4_Er phase still stably exists at the grain boundary after tensile test at 300 °C, which is sufficient to prove its thermal stability at high temperature and strengthening effect on grain boundaries. As stated in [47,48], in addition to the major precipitates, there is also an effect of Er-rich compounds at grain boundaries which hinder the motion of dislocations. Mei et al. also considered that Al_8_Cu_4_Y phases can make some contribution in improving the mechanical properties of the Al-Cu-Mg-Ag alloy at 300 °C because of their thermal stability [35]. Therefore, it can be inferred that the Al_8_Cu_4_Er phase plays a role in pinning the grain boundaries and preventing the slip of the grain boundaries, delaying the initiation, expansion, and connection of cracks during deformation at high temperature, thus improving the tensile strength of the Al-Cu-Mg-Ag alloy at high temperature.

## 5. Conclusions

In this work, the effect of minor Er additions on the microstructure and tensile properties of cast Al-Cu-Mg-Ag alloys were investigated and discussed. The main conclusions can be summarized as follows:(1)Al-Cu-Mg-Ag alloys with and without Er additions reach the peak aged condition after aging for 14 h, suggesting minor Er has little effect on the age-hardening response. However, the hardness value of the peak aged alloys slightly decreases from 141.8 ± 3.5 HV to 138.1 ± 2.7 HV and 137.9 ± 2.8 HV as Er increases from 0% to 0.2%.(2)No obvious grain refinement effect induced by Er additions can be found in as-cast Al-Cu-Mg-Ag alloys.(3)The precipitation of Ω plates is obviously restricted by Er while the denser precipitation of the θ′ phase is observed.(4)The tensile properties of Er-added Al-Cu-Mg-Ag alloys at room temperature degrade with Er addition, while the higher strength properties of Er-added alloys at 300 °C are ascribed to the Al_8_Cu_4_Er phase at the grain boundaries.(5)The Al_8_Cu_4_Er phase and (Mg, Ag, Er, V, Ti)-rich phase are detected in Er-added alloys, which cannot be dissolved in the Al-matrix after solution treatment and leads to an obvious brittle fracture of Al-Cu-Mg-Ag-Er alloys.

## Figures and Tables

**Figure 1 materials-14-04212-f001:**
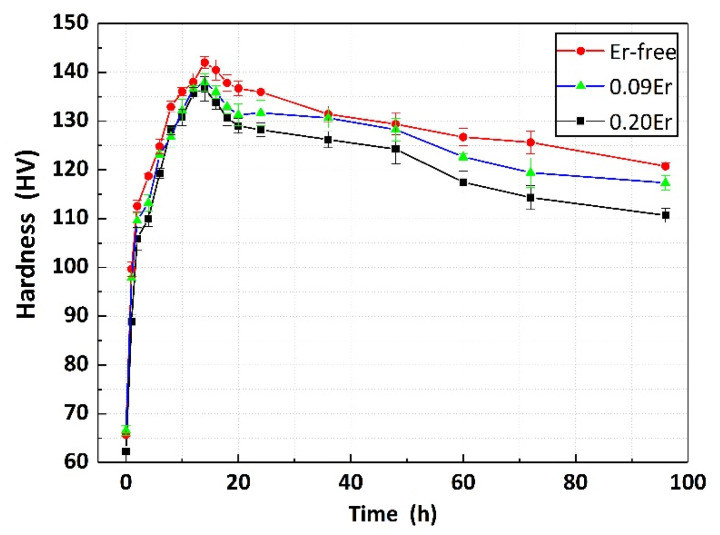
Age-hardening curves of studied alloys at 165 °C.

**Figure 2 materials-14-04212-f002:**
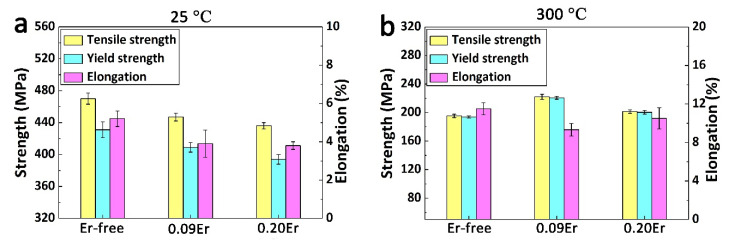
Tensile properties of studied alloys in T6 state at (**a**) 25 °C; (**b**) 300 °C.

**Figure 3 materials-14-04212-f003:**
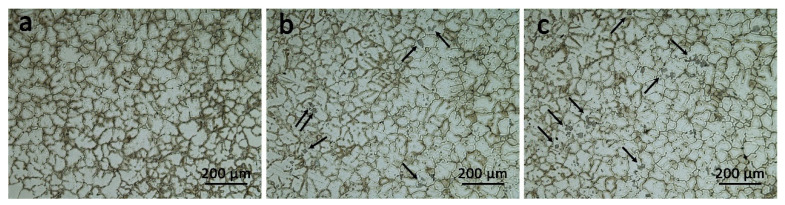
Metallographic images of three as-cast alloys (**a**) Er-free; (**b**) 0.09Er, (**c**) 0.20Er.

**Figure 4 materials-14-04212-f004:**
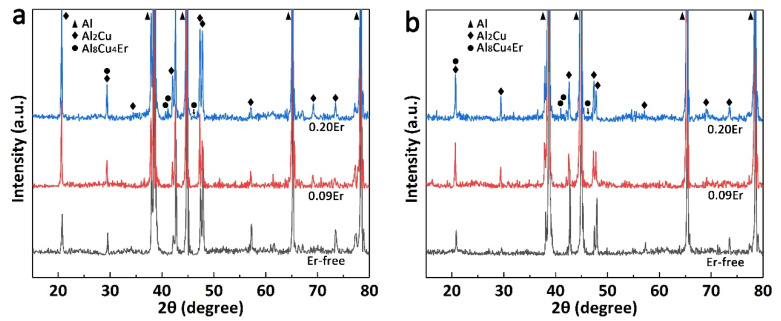
XRD patterns of three studied alloys of (**a**) as-cast state; (**b**) T6 state.

**Figure 5 materials-14-04212-f005:**
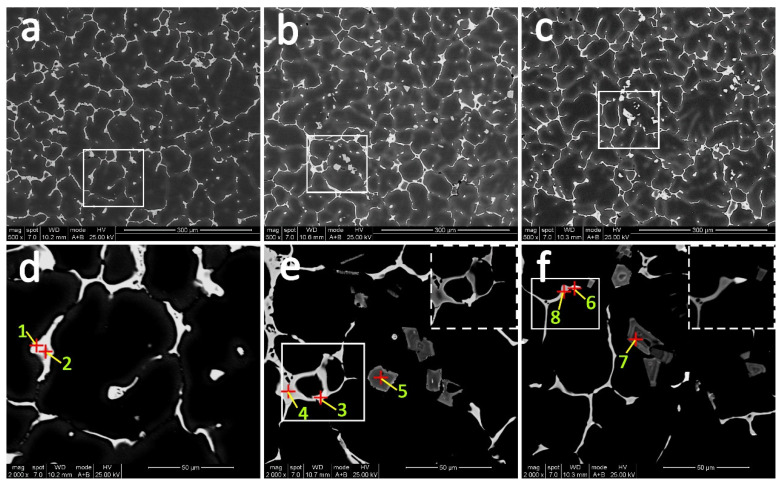
Backscattered electron (BSE) images of the as-cast alloys. (**a**,**d**) Er-free; (**b**,**e**) 0.09Er; (**c**,**f**) 0.20Er alloys.

**Figure 6 materials-14-04212-f006:**
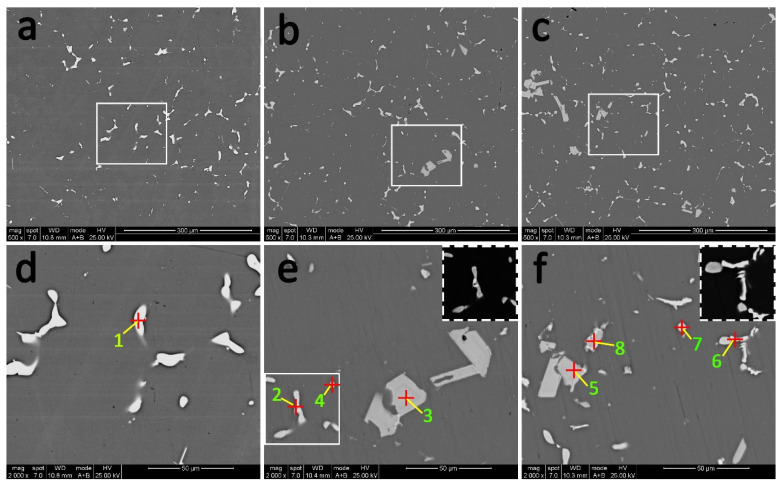
Backscattered electron (BSE) images of alloys after T6 heat treatment. (**a**,**d**) Er-free; (**b**,**e**) 0.09Er; (**c**,**f**) 0.20Er alloys.

**Figure 7 materials-14-04212-f007:**
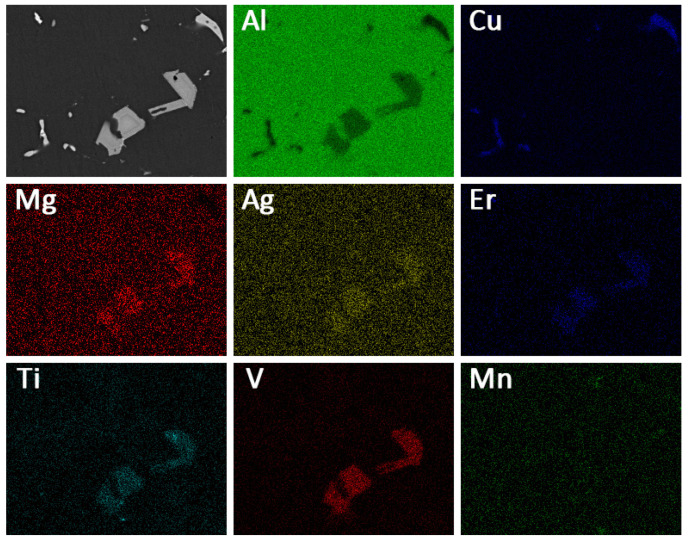
EDS mapping of selected region in Figure 6b.

**Figure 8 materials-14-04212-f008:**
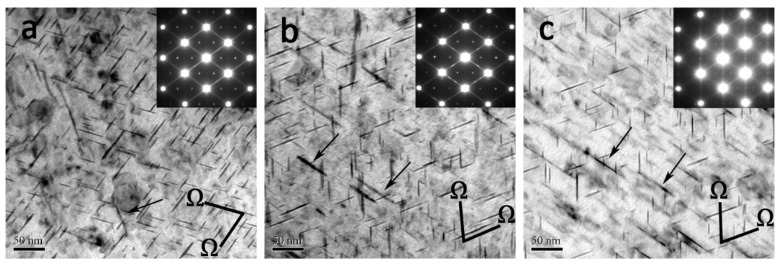
TEM images of studied alloys in T6 state and corresponding SAED patterns (near <110>_α_ direction). (**a**) Er-free; (**b**) 0.09Er; (**c**) 0.20Er.

**Figure 9 materials-14-04212-f009:**
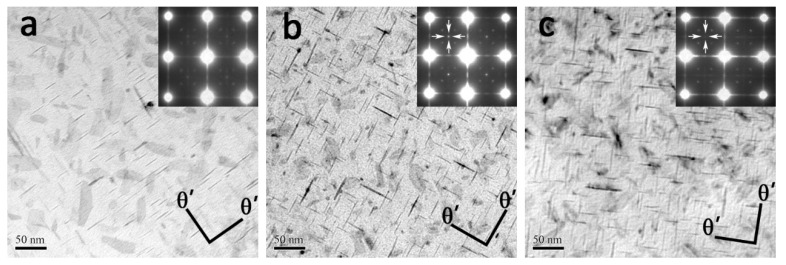
TEM images of studied alloys in T6 state and corresponding SAED patterns (near <100>_α_ direction). (**a**) Er-free; (**b**) 0.09Er; (**c**) 0.20Er.

**Figure 10 materials-14-04212-f010:**
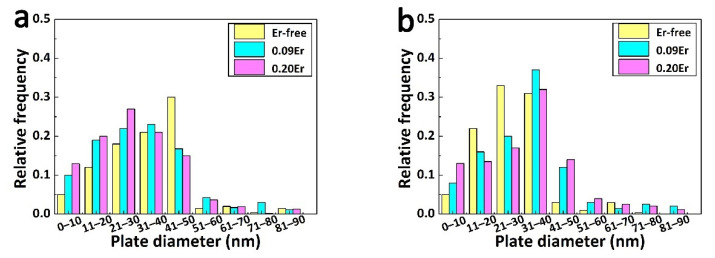
Statistical diagram of the distribution of the Ω and θ′ phases of studied alloys. (**a**) Ω phase; (**b**) θ′ phase.

**Figure 11 materials-14-04212-f011:**
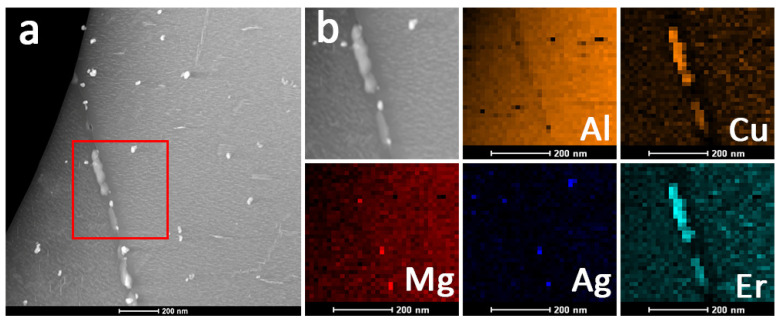
The (**a**) STEM-HAADF image and (**b**) corresponding EDS mapping of the microstructures of T6-tempered 0.09Er alloy after tensile test at 300 °C.

**Figure 12 materials-14-04212-f012:**
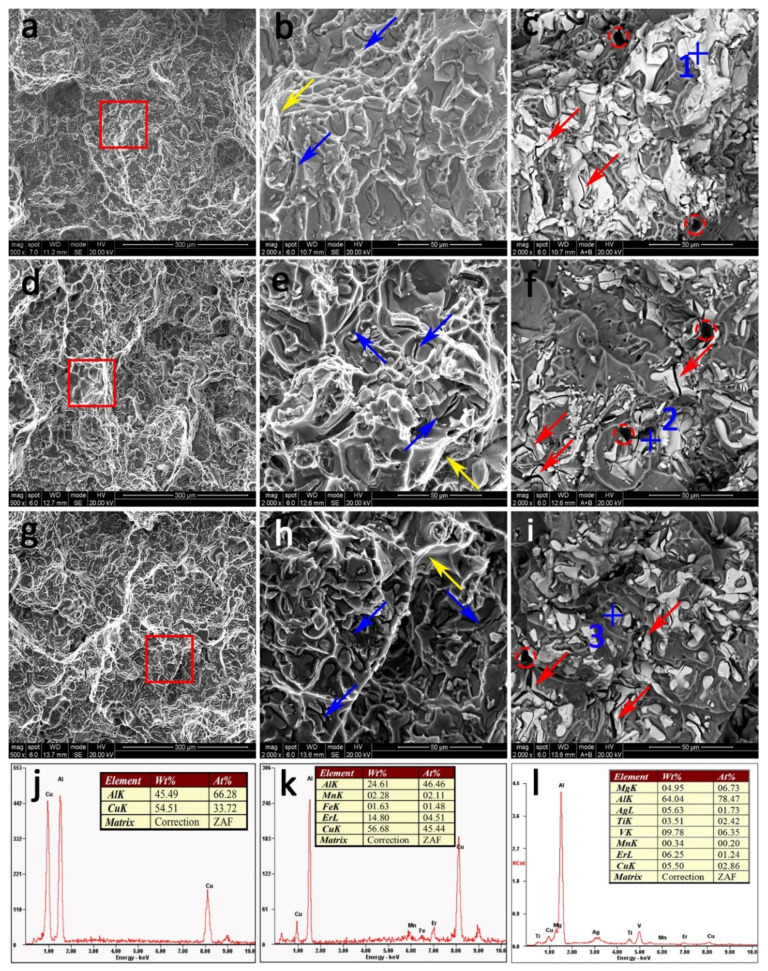
SEM images of the fracture surfaces of peak-aged alloys at room temperature in different magnifications. (**a**–**c**) Er-free; (**d**–**f**) 0.09Er; (**g**–**i**) 0.20Er; (**j**–**l**) EDS spectrum of points 1, 2 and 3.

**Table 1 materials-14-04212-t001:** Chemical composition of the studied Al-Cu-Mg-Ag-(Er) alloys (wt %).

Alloys	Cu	Mg	Ag	Mn	Ti	B	V	Er	Al
Er-free	5.97	0.32	0.81	0.30	0.15	0.02	0.16	-	Bal.
0.09Er	5.92	0.28	0.76	0.29	0.13	0.02	0.17	0.09	Bal.
0.20Er	5.95	0.33	0.72	0.29	0.13	0.02	0.16	0.20	Bal.

**Table 2 materials-14-04212-t002:** EDS analysis (wt %/at %) of red points marked in Figure 4.

Points	1	2	3	4	5	6	7	8
Al	52.60/72.32	60.83/77.86	36.90/62.23	44.74/65.59	49.60/84.36	56.08/77.67	68.40/83.53	47.33/67.91
Cu	47.40/27.68	28.34/15.41	46.42/33.23	55.26/34.41	4.65/2.39	27.20/16.00	3.94/2.04	52.67/32.09
Mg	-	-	-		2.34/3.15	-	1.81/2.45	
Ag	-	-	-		3.85/1.17	-	2.80/0.85	
Mn	-	3.39/2.13	-		-	5.69/3.87	-	
Ti	-	-	-		3.22/2.20	-	5.41/3.72	
Er	-	-	16.68/4.54		8.43/1.65	11.03/2.46	8.91/1.75	
V	-	-	-		7.91/5.08	-	8.73/5.65	
Fe	-	7.43/4.60	-		-	-	-	

**Table 3 materials-14-04212-t003:** EDS analysis (wt %/at %) of red points marked in Figure 5.

Points	1	2	3	4	5	6	7	8
Al	69.58/84.34	39.49/64.26	64.00/78.82	48.71/69.10	66.06/80.34	38.93/63.88	43.86/64.79	62.91/77.96
Cu	30.42/15.66	46.36/32.03	5.53/2.89	51.29/30.90	5.31/2.74	46.18/32.18	56.14/35.21	5.20/2.74
Mg	-	-	4.72/6.45	-	4.77/6.44	-	-	5.00/6.87
Ag	-	-	6.31/1.94	-	5.86/1.78	-	-	7.12/2.21
Mn	-	-	-	-	-	-	-	0.27/0.16
Ti	-	-	2.78/1.93	-	2.66/1.82	-	-	3.14/2.19
Er	-	14.15/3.71	6.39/1.72	-	6.72/1.32	14.89/3.94	-	6.30/1.26
V	-	-	10.27/6.70	-	8.62/5.55	-	-	10.06/6.60

**Table 4 materials-14-04212-t004:** Quantitative analysis of TEM results of the Ω and θ′ phases in cast Al-Cu-Mg-Ag-(Er) alloys in peak-aged state.

Alloys	Ω Phase			θ′ Phase		
	Plate Diameter (nm)	Plate Thickness (nm)	Number Density (×10^3^/μm^3^)	Plate Diameter (nm)	Plate Thickness (nm)	Number Density (×10^3^/μm^3^)
Er-free	42.3 ± 12.4	2.6 ± 1.2	27.6 ± 10.3	32.7 ± 10.1	2.5 ± 1.0	2.7 ± 2.1
0.09Er	37.8 ± 9.5	2.3 ± 1.1	21.1 ± 9.1	36.8 ± 15.4	2.3 ± 0.9	4.3 ± 3.1
0.20Er	37.5 ± 6.3	2.4 ± 1.2	19.4 ± 1.1	35.4 ± 16.3	2.6 ± 0.8	4.1 ± 6.7

## Data Availability

The raw/processed data required to reproduce these findings cannot be shared at this time due to technical or time limitations.

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
