# Peer review of "Effect of Minor Er Additions on the Microstructures and Mechanical Properties of Cast Al-Cu-Mg-Ag Alloys"

_materials, 2021, doi:10.3390/ma14154212_

Round 1
Reviewer 1 Report
The article is interesting and has utility values from the point of view of the research tools used, but it does not contain significant and new information about these materials. A small addition of Er has no effect on the precipitation hardening of the cast Al-Cu-Mg-Ag alloys and has no grain refinement effect. The mechanical properties deteriorate. Most likely, there is not enough Er supplement when compared with [12]. In my opinion, the tests should be repeated with a higher Er content in cast Al-Cu-Mg-Ag alloys. Therefore, I suggest a major revision.
While reading the article, you may come across the following shortcomings:
- there is no justification why the influence of such a small addition of Er on the properties of cast Al-Cu-Mg-Ag alloys was investigated;
- the size of the Er supplement weight appears to be completely random and has not been substantiated;
- in chapter 1, at the end, it should be stated what the purpose of the work was;
- no units in table 1, for example: concentration [wt.%];
- no item 3.6.3. or the numbering should be changed.
Reviewer 2 Report
Dear authors, the work presented in this manuscript is very interesting and the experimental methodology is important. However, I think the manuscript needs to be improved. Especially in the introduction and in the analysis of the state of the art. On the other hand, the work reflects “doubts” in references to the influence of the microstructure in the as cast state and the dissolution treatment carried out. I recommend that these "doubts" be clarified to improve the manuscript.
The introduction indicates how the addition of a series of rare earths influences the properties of various aluminum alloys with hardening by aging. The alloy that is the object of this study is an Al-Cu-Mg-Ag alloy, however, in the introduction, the authors comment on other Al-alloys. For example, Al-Mg-Si; Al-Cu-Sc; Al-Zn-Mg, etc. On the other hand, the structure of the introduction seems messy. I recommend that the authors improve the introduction. To do this, I recommend that the authors take into account the following considerations:
- Main characteristics and applications of the Al-Cu-Mg-Ag alloys that are the object of this study. Based on these applications, what is the problem to be solved?
- State of the art, with respect to Al-Cu-Mg-Ag alloys and with respect to the applications described above
- What new and relevant results have been achieved.
In the experimental part, the authors should explain the dissolution treatment used:
- In the as cast state, did the microstructure present dendritic segregation (non-equilibrium solidification)?
- Why did the authors select 530 ºC? Can the authors confirm that they have not exceeded the true solidus temperature?
- Why do the authors select a treatment duration as long as 12 hours?
Regarding the results: In the micrographs in the as cast state (Figure 3 and Figure 5) it is observed that the microstructure presents dendritic segregation. Are the authors sure that the temperature of 530 ºC was not excessive? Please clarify
Reviewer 3 Report
Some of the figure have a wrong number. I highlited it.
From my point of view, it will be better merging results and discussion to one chapter. The editing of text part make beter the orientation in the text and understandability of the results.
SEM analysis is discussed directly in the results, in contrast, TEM observations are fragmented and the text is both in the results and in the discussion.

Round 2
Reviewer 1 Report
The article is interesting and has utility values from the point of view of the research tools used. The Authors' explanations do not dispel my doubts, but I understand that repeating the measurements would be very time-consuming. The Authors summarized in an abstract why the tests were conducted with a low Er content. The explanations are compelling.
While reading the article, you may come across the following shortcomings:
1. the formatting of Figures 2-9 and expressions (1-3) in the text should be improved.